# A Portable Intuitive Haptic Device on a Desk for User-Friendly Teleoperation of a Cable-Driven Parallel Robot

**Jae-Hyun Park** [1,†], **Min-Cheol Kim** [1,†], **Ralf Böhl** [2], **Sebastian Alexander Gommel** [2], **Eui-Sun Kim** [3], **Eunpyo Choi** [1,3], **Jong-Oh Park** [3] and **Chang-Sei Kim** [1,3,*] 

1   School of Mechanical Engineering, Chonnam National University, Gwangju 61186, Korea; Jhpark2068@naver.com (J.-H.P.); kmc100291@gmail.com (M.-C.K.); eunpyochoi@jnu.ac.kr (E.C.)
2   School of Manufacturing Engineering, University of Stuttgart, 70049 Stuttgart, Germany; sebastiangommel@hotmail.com
3   Korea Institute of Medical Microrobotics, Gwangju 61011, Korea; kes@kimiro.re.kr (E.-S.K.); jop@kimiro.re.kr (J.-O.P.)
*   Correspondence: ckim@jnu.ac.kr; Tel.: +82-62-530-5260
†   These authors contributed equally.

**Abstract:** This paper presents a compact-sized haptic device based on a cable-driven parallel robot (CDPR) mechanism for teleoperation. CDPRs characteristically have large workspaces and lightweight actuators. An intuitive and user-friendly remote control has not yet been achieved, owing to the unfamiliar multiple-cable configuration of CDPRs. To address this, we constructed a portable compact-sized CDPR with the same configuration as that of a larger fully constrained slave CDPR. The haptic device is controlled by an admittance control for stiffness adjustment and implemented in an embedded microprocessor-based controller for easy installation on an operator's desk. To validate the performance of the device, we constructed an experimental teleoperation setup by using the prototyped portable CDPR as a master and larger-size CDPR as a slave robot. Experimental results showed that a human operator can successfully control the master device from a remote site and synchronized motion between the master and slave device was performed. Moreover, the user-friendly teleoperation could intuitively address situations at a remote site and provide an operator with realistic force during the motion of the slave CDPR.

**Keywords:** cable-driven parallel robot; haptic device; haptic interaction; intuitive teleoperation

## 1. Introduction

A cable-driven parallel robot (CDPR) is a special type of parallel robot that is actuated by elastic lightweight cable links instead of rigid links, which are generally used in other robot mechanisms and automated systems. The main components of a CDPR are cables, winches, motors, pulleys, an end-effector, etc. By virtue of the elastic cable links, a CDPR offers several advantages, such as a large workspace, highly dynamic features, and higher payloads compared with existing parallel robots [1–3]. By maximizing these advantages, CDPRs are important in industrial and robotics research where highly dynamic features, accurate position control, and high payload performances are required in a large workspace [4]. For example, rehabilitation [5–7], large-scale 3D printing [8–10], localization [11], and a CDPR human-like robotic arm [12] are representative recent developments.

However, an intuitive and user-friendly remote control has not yet been achieved due to the unfamiliar multiple-cable configuration of CDPRs. As a master device to control a CDPR, a commercialized compact haptic device can be useful, as haptic devices have widely been used for various applications, such as surgical robot systems [13,14], virtual haptic systems [15,16], and teleoperation [17,18]. Importantly, especially in dangerous workplaces, intuitive remote motion control is an essential part of user-commanded operation. Therefore, the intuitive remote control proposed in our previous research [19]

and vision-based control [20] for a CDPR have been addressed. Typical haptic devices are composed of serial [21] and parallel [22] types of rigid actuators; the motion of an end-effector is controlled by users through joysticks, knobs, or exoskeletons. Although various types of haptic devices are available, the intuitive motion discrepancy between haptic devices and CDPR end-effectors still exists due to the complex configuration of CDPRs. Their operation range is also often limited by being applied to an elastic cable link of a CDPR because of the rigid links of conventional haptic devices. Exoskeleton haptic devices are also difficult for remote users to use because of inconveniences while wearing them and weak controllability. Therefore, portability and easy installation on an operator's desk or bench, in addition to intuitive operation, will enhance the user-friendly remote controllability of a CDPR.

Previously, research on utilizing parallel-type haptic devices has been conducted. [23] developed a prototype cable-driven haptic device operated by 7 DOF. Redundant parallel-mechanism-based haptic devices, named DELTA-R, were developed as prototypes for virtual-reality applications [24]. Yang Ho W et al. developed a haptic-interaction system using a MINI IPAnema 3, manufactured by Fraunhofer IPA, based on the admittance control algorithm, and proved its performance through experimental verification [25]. However, a compact-size haptic device configured by a CDPR mechanism with an embedded microcontroller has not been achieved for practical application.

In this study, we constructed a new type of CDPR-configured haptic device for user-friendly teleoperation. The proposed system can maximize the advantages of remote CDPR operation with easy and intuitive motion control. The proposed method can also provide a practical solution for teleoperation systems requiring a haptic device that covers a large workspace, but is compact enough to ensure security and convenience for the operator. The haptic device of the proposed system can help operators easily control a CDPR by allowing them to feel as though they are actually holding the object. Thus, they can flexibly address situations in remote sites even though objects are not observable at their location.

For practical implementation, we designed and prototyped a compact CDPR. To achieve an on-desk-size system, we used a microprocessor that enabled the portable haptic device to be controlled independently with a PLC-based computing system. For the stiffness adjustment, the admittance control algorithm was implemented for haptic interaction in the master device. The position commands were generated by using the movement of the master haptic device and transferred to the slave robot in real time though USB communication. An adjustable scale factor was applied to compensate for the size differences between the master and slave devices. We constructed the teleoperation system using the developed haptic CDPR as the master device and another as the slave device.

This paper is organized as follows: The theoretical background of the developed haptic device, including the kinematics, dynamics, and admittance control, is explained in Section 2. The practical design and fabrication of the compact CDPR hardware and microprocessor-based controller, as well as the proposed control algorithm and the configuration of teleoperation, are shown in Section 3. In Section 4, the experimental setup, results, and analysis are provided. Finally, we present a discussion of the results and future work.

## 2. Kinematics, Dynamics, and Control of the Haptic CDPR

### 2.1. Kinematic and Wrench Force of the CDPR

The kinematic geometry of the $i^{th}$ cable in a CDPR is depicted in Figure 1. The geometric parameter $A_i$ represents the connecting point between the cable and the base frame, and $B_i$ is the connecting point between the cable and the end-effector. Variables $a_i$ and $b_i$ are two constant vectors on the base coordinate $\{G\}$ and the end-effector coordinate $\{P\}$, respectively.

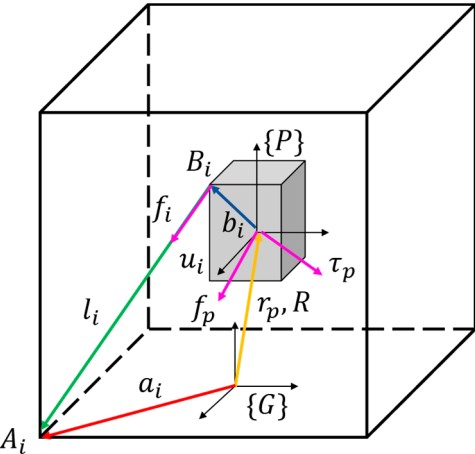

**Figure 1.** Kinematic geometry of the $i^{th}$ cable in CDPR.

The inverse kinematics of the CDPR used in this study compute the cable length in the joint space from the given end-effector posture in the Cartesian space. Based on the closure vector loop of the CDPR, the cable length vector can be calculated, where $l_i$ represents the vector for the $i^{th}$ cable, and $r_p$ and $R$ are the position vector and rotation matrix of the end-effector with respect to the base coordinate $\{G\}$, respectively:

$$l_i = a_i - r_p - Rb_i \tag{1}$$

The length of each cable $l_i$ is obtained by the 2-norm of (1) as follows:

$$l_i = \sqrt{\left[a_i - r_p - Rb_i\right]^T \cdot \left[a_i - r_p - Rb_i\right]} \tag{2}$$

To calculate the wrench force applied to the end-effector from the measured tension of each cable, the force and torque equilibria are applied as follows:

$$\sum_{i=1}^{N} f_i + f_p = 0, \text{ and } \sum_{i=1}^{N}(Rb_i \times f_i) + \tau_p = 0 \tag{3}$$

where $N$ is the number of cables in CDPR, $f_i$ is the tension vector of the $i^{th}$ cable, and $f_p$ and $\tau_p$ are the external force and torque applied to the end-effector, respectively. Then, by considering the unit vector that represents the direction of the tension $u_i = l_i / \|l_i\|$, we can obtain a closed form of dynamics for the CDPR as follows:

$$\underbrace{\begin{pmatrix} u_i & \cdots & u_N \\ Rb_i \times u_i & \cdots & Rb_N \times u_N \end{pmatrix}}_{A^T} \begin{pmatrix} f_1 \\ \vdots \\ f_N \end{pmatrix} + \underbrace{\begin{pmatrix} f_p \\ \tau_p \end{pmatrix}}_{w} = 0 \tag{4}$$

Finally, the simplified form of (4) and the force limitation boundary can be written as follows:

$$A^T f + w = 0, f_{max} \ge f \ge f_{min} > 0 \tag{5}$$

where $w$ is the external wrench force, $A^T$ is the structure matrix, and $f$ is the cable tension.

### 2.2. Admittance Control

Ideally, the robot becomes an admittance object that is powered by input and outputs to a position. When interacting with a human, the human acts as an impedance object that outputs force. Reversely, this can be used for adjustable stiffness control for an object, called an admittance control [26,27]. Therefore, we used the admittance control algorithm, which can estimate position set points through a spring-mass-damper (SMD) system model

using the external force applied to an object. A basic 1 DOF SMD system comprises three dynamic parameters: mass *M*, damping coefficient *C*, and stiffness coefficient *K*. This system can be represented using the following $2^{nd}$-order differential equation:

$$M\ddot{x} + C\dot{x} + Kx = F(t) \tag{6}$$

where $F(t)$ is the external force applied to the system; $x$ is the position; $\dot{x}$ and $\ddot{x}$ are, respectively, the velocity and acceleration, which are the derivatives of the position with respect to time. To obtain the real solution of this $2^{nd}$-order differential equation, the Laplace transformation method [28,29] is used in this proposed admittance control as:

$$x(s) = \frac{1}{ms^2 + cs + k}f(s) \tag{7}$$

where $x(s)$ is the displacement according to the external force $f(s)$. The variables *M*, *C*, and *K* of the transfer function are the dynamic parameters of the SMD system.

Practically, to implement (7) into an embedded controller we used the $4^{th}$-order Runge-Kutta method. However, there are several numerical analysis methods, such as the Euler and Runge-Kutta methods, that can minimize the computation time when applying the mathematical model to a real system. The form of the $4^{th}$-order Runge-Kutta method is derived as follows:

$$X_{n+1} = X_n + \frac{h}{6}(k_1 + 2k_2 + 2k_3 + k_4) \tag{8}$$

The extended SMD system for the 6 DOF space can be derived as follows: This system comprises three dynamic parameters that describe the end-effector dynamics in the wrench space: mass and inertia *I*, damping *D*, and stiffness coefficient *C*.

$$I\ddot{r} + D\dot{r} + Cr = w(t) \tag{9}$$

where $w(t)$ is the applied external wrench force that represents the external force and torque; and $r$, $\dot{r}$, and $\ddot{r}$ denote the position, velocity, and acceleration, respectively. So, SMD parameter adjustment in (7) results in the impedance changes in (9); thus, we can adjust the force feedback to the operator's hand.

## 3. Design and Fabrication of the Haptic CDPR

### 3.1. Hardware Design

The specifications of the proposed haptic device's design objectives are shown in Table 1. The velocity of the end-effector was arbitrarily decided by assuming that the end-effector moves the maximum distance in the workspace of 207.86 mm in 2 s through workspace analysis. In addition, the angle of force measurement was also considered to ensure the precise measurement of the tension in the cable.

**Table 1.** Established specifications of the proposed haptic device.

| Objective | Specification |
|---|---|
| DOF | 6 DOF |
| Robot Size | 240 × 240 × 240 mm |
| Workspace | 120 × 120 × 120 mm |
| External Force | 10 N |
| Velocity | 0.1 m/s |
| Angle of force measurement | 90° |

The design and the developed prototype of the main frame are shown in Figure 2. The proposed device mainly consists of pulleys, motors (IG32GM 256PPR 09 TYPE), force sensors (CZL 635), and winches, which are integrated with pulleys to enable end-effector movement through winding or unwinding of the cable. There are other remaining

components and an aluminum frame for mounting components. When designing the proposed haptic device, we specifically considered the angle of force measurement of 90 ° for precise tension measurement in the cable, preventing tension measurement errors due to the divided force components. Figure 3 shows the estimated workspace of the developed haptic device.

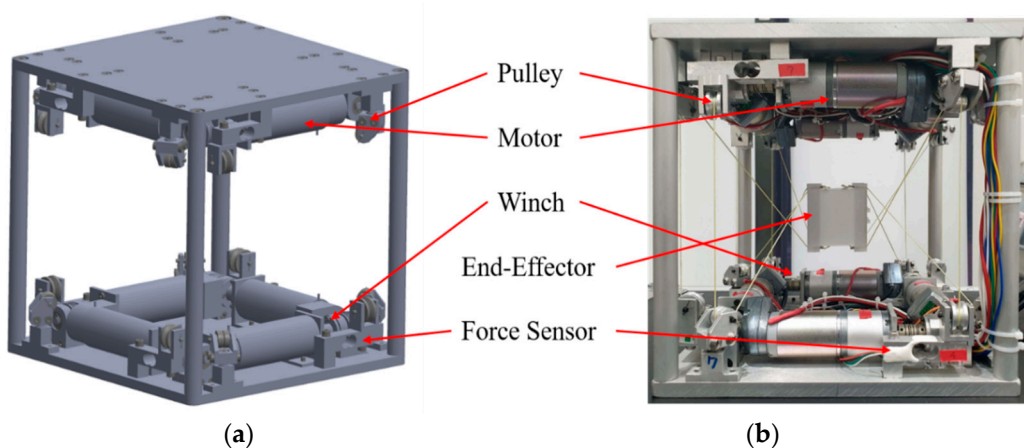

**Figure 2.** Designed and fabricated models of the main part of the system: (**a**) designed model of the main system; (**b**) fabricated model of the main system.

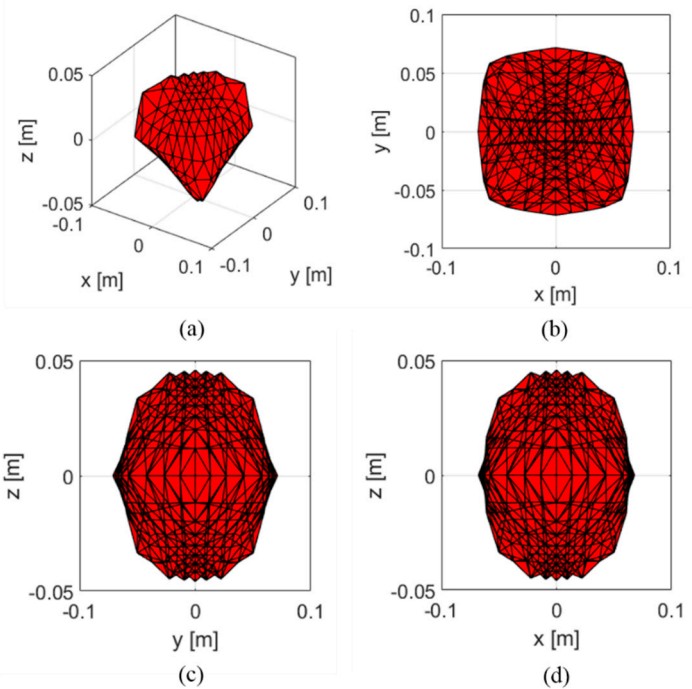

**Figure 3.** Estimated workspace boundary of the proposed system: (**a**) Workspace in 3D space; (**b**) Workspace in X-Y plane; (**c**) Workspace in Y-Z plane; (**d**) Workspace in X-Z plane.

We used a closed-form force distribution approach to estimate the workspace by using Equation (5). The force vector $f$ is divided into $f_m = (f_{max} + f_{min})/2$ and $f_v$, which represent the mean feasible force distribution and the arbitrary force vector, respectively [30]. Through the estimation, the calculated workspace boundary was $X = \{-0.068, 0.068\}$, $Y = \{-0.071, 0.071\}$, and $Z = \{-0.046, 0.046\}$. The selected tension boundary of the cable was 0 to 50 N, considering the payload of the force sensor, and the geometrical parameters were obtained as listed in Table 2.

**Table 2.** Geometric parameters of the haptic device.

| Cable No. | Position (mm) | |
| :---: | :---: | :---: |
| | Pulleys ($A_i$) | Edge of End Effector ($B_i$) |
| 1 | $[-110, -80, 72.5]$ | $[-20.5, -24.5, -27.5]$ |
| 2 | $[80, -110, 72.5]$ | $[20.5, -24.5, -27.5]$ |
| 3 | $[110, 80, 72.5]$ | $[20.5, 24.5, -27.5]$ |
| 4 | $[-80, 110, 72.5]$ | $[-20.5, 24.5, -27.5]$ |
| 5 | $[-80, -110, -72.5]$ | $[-20.5, 24.5, 27.5]$ |
| 6 | $[110, -80, -72.5]$ | $[20.5, -24.5, 27.5]$ |
| 7 | $[80, 110, -72.5]$ | $[20.5, 24.5, 27.5]$ |
| 8 | $[-110, 80, -72.5]$ | $[-20.5, 24.5, 27.5]$ |

### 3.2. Embedded Controller

To build a small-sized control system in our development, Nucleo microprocessors were selected after considering compactness, performance, and accessibility. To increase the computation time, we used two microprocessors based on parallel processing communication. In the control part, the Nucleo-64 obtains the signals from the force sensor through the amplifier and the Nucleo-144 drives the motor. The force sensor signals from the Nucleo-64 are transferred to the Nucleo-144 through serial communication. When selecting the digital/analog pins in the microprocessor, the number of channels in the microprocessor was also considered, preventing collisions between the signals. Figure 4 shows the controller components of the developed haptic device.

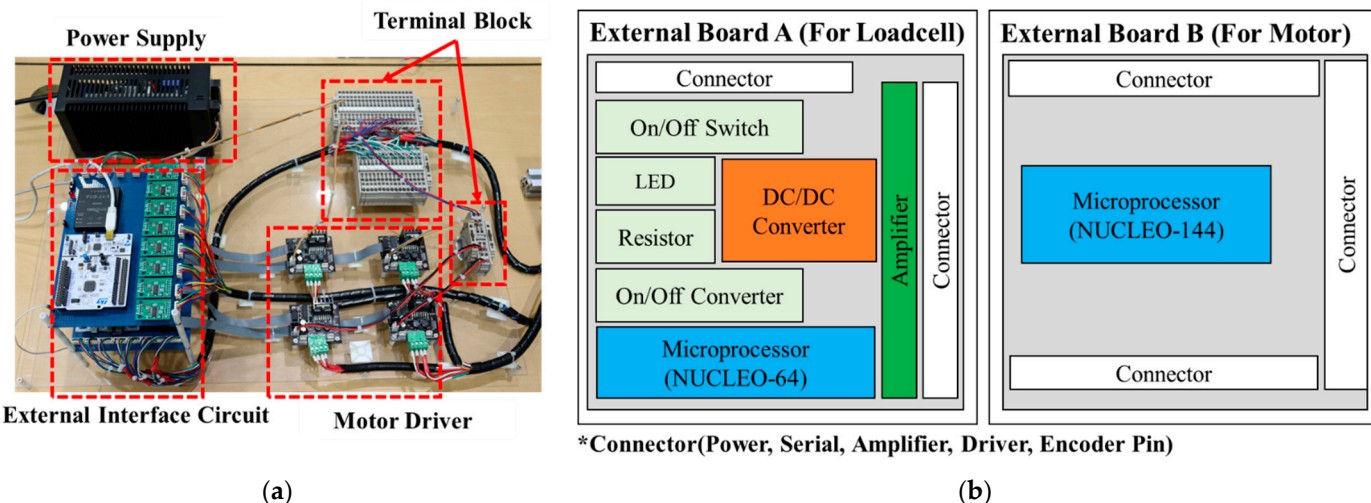

(**a**)  (**b**)

**Figure 4.** External board setup part of developed system: (**a**) controller part of the developed haptic device; (**b**) schematic of the external board.

The other components for the controller are the power supply (VSF220-24) to the device, the drivers (AM-DC2-2D) for amplifying low current signals from the microprocessor and producing a high current signal that can control and drive the motor, the terminal blocks, and a separately fabricated external board for convenient troubleshooting. The external board comprises the following components: an on/off switch for the device, a resistor for preventing overcurrent from the device, a DC/DC converter (PS15-24-5) for converting high voltage from 24 to 5 V (to power the components that need 5 V power, such as microprocessors, encoders, drivers, and amplifiers), microprocessors that communicate signals between other components, amplifiers (HX711) for amplifying low current signals from the force sensor and producing high current signals to enable the microprocessor to read the signals, and connectors for conveniently connecting the signals.

Based on the prototyped microprocessor and sensors, the haptic CDPR control algorithm was developed including robot kinematics, inverse kinematics, force control, and admittance control. Figure 5 depicts a block diagram of the proposed control algorithm for the developed haptic device. For the given desired movement of the end-effector by an operator, the adjusted force related to the stiffness and haptic feeling of the operator can be provided to the operator with motion command generation to the slave CDPR.

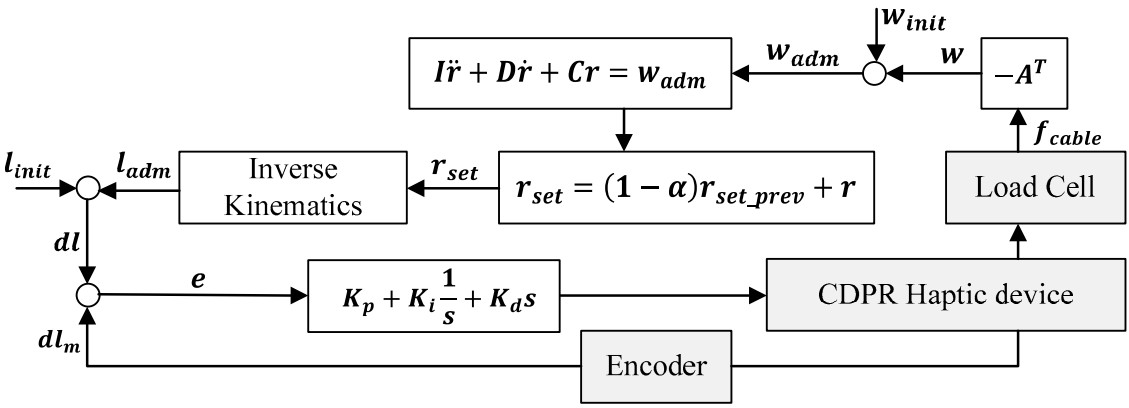

**Figure 5.** A block diagram of the proposed control algorithm for a haptic CDPR.

First, if the operator applies an external wrench force $w$ to the end-effector, $w$ is calculated by multiplying the structure matrix $-A^T$ with the cable tensions $f_{cable}$ measured from the force sensors, as described in Section 2.1. Second, the wrench force to implement the admittance control $w_{adm}$ can be derived using the initially calculated wrench force ($w_{adm} = w - w_{init}$) in order to eliminate the initial measurement error of the force sensor. Then, the target position $r_{set}$ for the admittance control can be derived based on the SPD system and additional data filtering using a $1^{st}$-order low-pass filter with a cut-off frequency of 1 Hz. Lastly, the transformed target cable length for admittance control $I_{adm}$ on the joint space is obtained from the inverse kinematics and the error between $I_{adm}$ and $I_{init}$ is compared to the measured value from the encoder to be an input to the motors via the PID controller. The respective gains for the SPD system ($I = 3$, $D = 50I$, $C = 183I$) and the PID controller ($K_p = 4.88 \times 10^{-4}$, $K_i = 0.08K_p$, $K_d = 0.024K_p$) were determined by trial and error. The developed control system was implemented using C++ through the Mbed compiler, provided by STMElectronics, Geneva, Switzerland with a sampling time of 15 ms.

### 3.3. Teleoperation Scheme

Figure 6 shows a schematic diagram of the teleoperation system. USB communication was chosen for our teleoperation system as it allows for simple communication via a single wire. The developed haptic device was used as the master device for the teleoperation and a CDPR called the MINI IPAnema 3 (Fraunhofer IPA, Stuttgart, Germany) as the slave device.

The PLC language is Structured text, and the based robot operating program is TwinCAT3 for the MINI IPAnema3, which can communicate with a C#-based program via ADS. This ADS allows all TwinCAT3 servers and client programs to exchange commands and data, even when they are written using different languages. Therefore, for data communication with the Nucleo microprocessor and the MINI IPAnema 3, the C#-based program was separately developed using Visual Studio. Via this data communication, the calculated position was transferred from the master to the slave device.

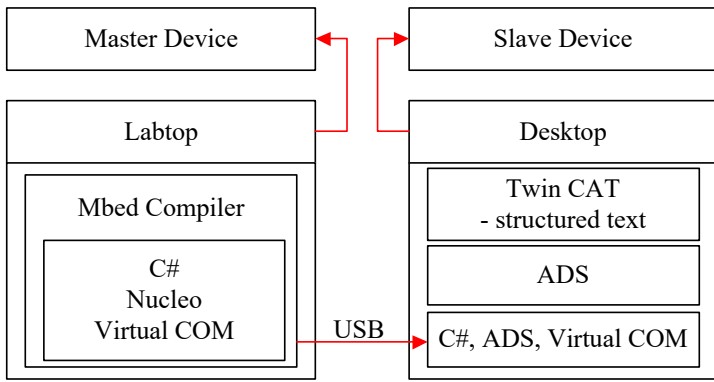

**Figure 6.** Schematic diagram of the proposed teleoperation system.

## 4. Experimental Setup and Results

We validated the performance of the developed haptic device with three experiments in terms of position, admittance control, and teleoperation. We built a remote-control experimental setup using the prototyped haptic device and MINI IPAnema3 CDPR.

### 4.1. Experimental Setup

To evaluate the performance of the proposed haptic device, we conducted an experiment using the prototyped CDPR haptic device and MINI IPAnema 3. Figure 7a shows the experimental setup for measuring the positional accuracy of the developed device. The setup comprised an optical tracking system (OTS) that can measure and store the position and rotation of its marker with a frequency of 60 Hz, an end-effector, an attached marker, and a formulated plate for moving the end-effector to the home position. The positional accuracy was evaluated by comparing the desired and measured positions. Figure 7b shows the experimental setup for evaluating the applied admittance control algorithm and the teleoperation system.

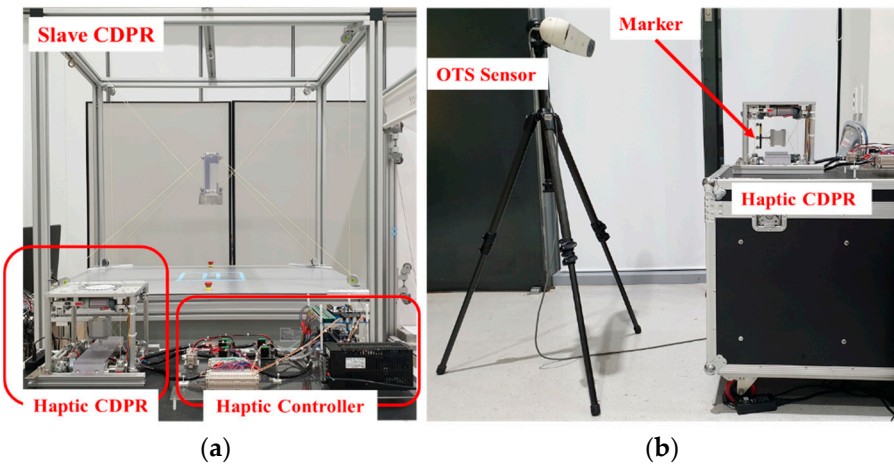

**Figure 7.** Experimental setup: (**a**) experimental setup for evaluating the positional accuracy of the master device; (**b**) experimental setup for evaluating the applied admittance control algorithm and the teleoperation system.

The setup for evaluating the admittance control consisted of force sensors located on the prototype device, weights to calibrate the force sensor, and another plate to move the end-effector to the certain path. The admittance control was evaluated by comparing the position and force outputs of the different SMD with respect to the circular path. To validate the performance for teleoperation, the developed device was connected to the

MINI IPAnema 3 via USB, and it was compared with the end-effector positions of the master and the slave device.

### 4.2. Validation of the Position Control

The motion control accuracy of the proposed device was evaluated as shown in Figure 8. The desired path was given as a rectangular path with the dimensions of $50 \times 50$ mm on the $X, Y$ axis and an additional linear path of 20 mm on the $Z$ axis. The velocity of the desired point along the path was set to 10 mm/s. For a smooth and precise movement of the end-effector, a cubic velocity profile was used. The desired position of the end-effector on the path is shown in Figure 8a. The movement of the end-effector started and ended at the home position of {0, 0, 0}. Figure 8b shows the desired and measured positions and the position error in each axis. The maximum absolute value for the position error was measured as 3.7, 2.54, and 1.45 mm for the $X, Y,$ and $Z$ axis, respectively. The average standard deviation and RMS of the measured errors are shown in Table 3.

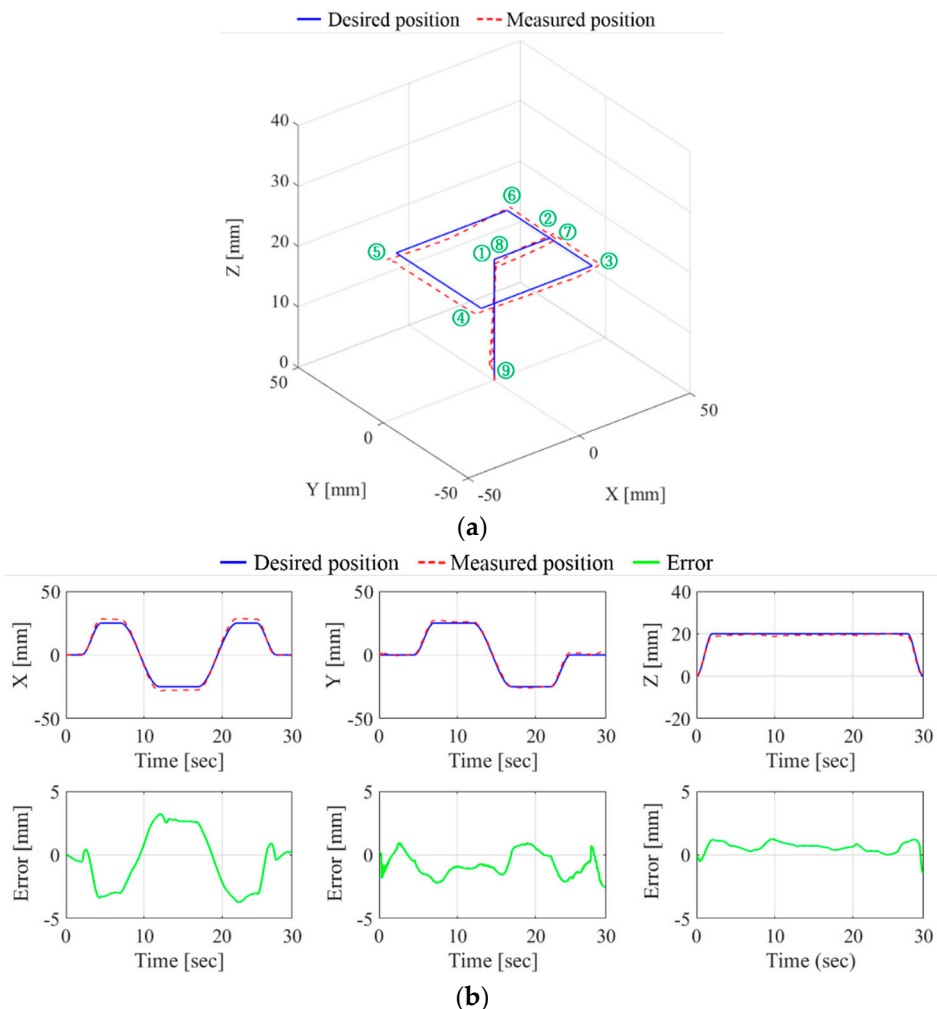

**Figure 8.** Comparison of the desired position and measured position in (**a**) 3D space and (**b**) 1D space.

**Table 3.** Measured error of the EE on the trajectory.

| Parameter | Mean (mm) | SD (mm) | RMS (mm) |
| --- | --- | --- | --- |
| $X$ axis | −0.4253 | 2.2046 | 2.2446 |
| $Y$ axis | −0.7673 | 0.9058 | 1.1869 |
| $Z$ axis | 0.6157 | 0.4005 | 0.7344 |
| Magnitude | 1.0718 | 2.4169 | 2.6432 |

*4.3. Validation of the Admittance Control*

We validated the applied admittance control algorithm by comparing different virtual systems with high and low values of dynamic parameters. The high dynamic parameters used in the equation of the SMD system were set to $I_H = 3$, $D_H = 100I_H$, and $C_H = 557I_H$. The low dynamic parameters were $I_L = 3$, $D_L = 50I_L$, and $C_L = 183I_L$.

In this experiment, the developed device was controlled manually by the human operator. Prior to starting the experiment, the force sensor was calibrated precisely by hanging weights ranging from 0 to 5 kg, in increments of 0.5 kg, on each cable according to its working load.

A circular trajectory with a radius of 20 mm on the $X$, $Y$ axis and an additional linear trajectory of 8 mm on the $Z$ axis was used. Figure 9 shows the comparison of the end-effector position and applied wrench force between the two systems. The solid blue line and the orange line represent the experimental result with high and low dynamic parameters, respectively. The experimental results indicated a considerable difference in the magnitudes of the respective wrench forces used in the two systems compared with the differences between the respective end-effector positions. Slight oscillations occurred in the presence of the external force because of friction due to the cable and inaccurate geometric parameters. Even where an external force did not exist, slight oscillations also occurred due to the gravitational force on the $Z$ axis.

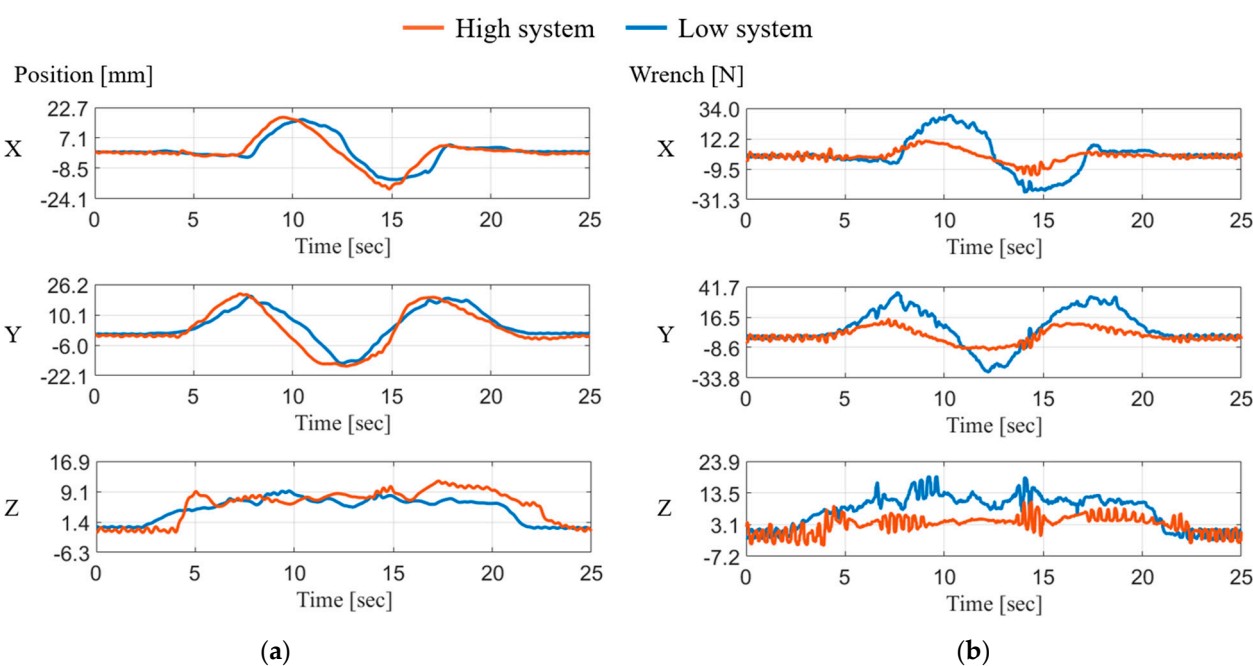

**Figure 9.** Comparison of the (**a**) position value and (**b**) applied wrench force according to the different virtual systems.

*4.4. Evaluation of the Proposed Teleoperation System*

The operator arbitrarily controlled the master device by directly applying the external wrench force to the end-effector, and its positional command was used as the input for the

slave device through data communication. Furthermore, the positions of the master and the slave device were simultaneously measured using the OTS in real time.

Figure 10 shows a comparison of results for the end-effector positions of the master and slave devices in Cartesian space. The maximum absolute values of the position tracking error between the master and slave device were measured as 3.21, 1.38, and 2.54 mm for the $X$, $Y$, and $Z$ axis, respectively. The average standard deviation and RMS of the measured errors are shown in Table 4. The measured time delay of the developed teleoperation system was 35 ms. From the comparison, it is evident that the slave CDPR smoothly followed the master CDPR during teleoperation. However, the magnitude of each position was slightly different because of a geometric error caused by neglecting the size of the pulley in the kinematic equation used in order to minimize the computation time and because of the designed scale factor regarding the spatial workspace volume.

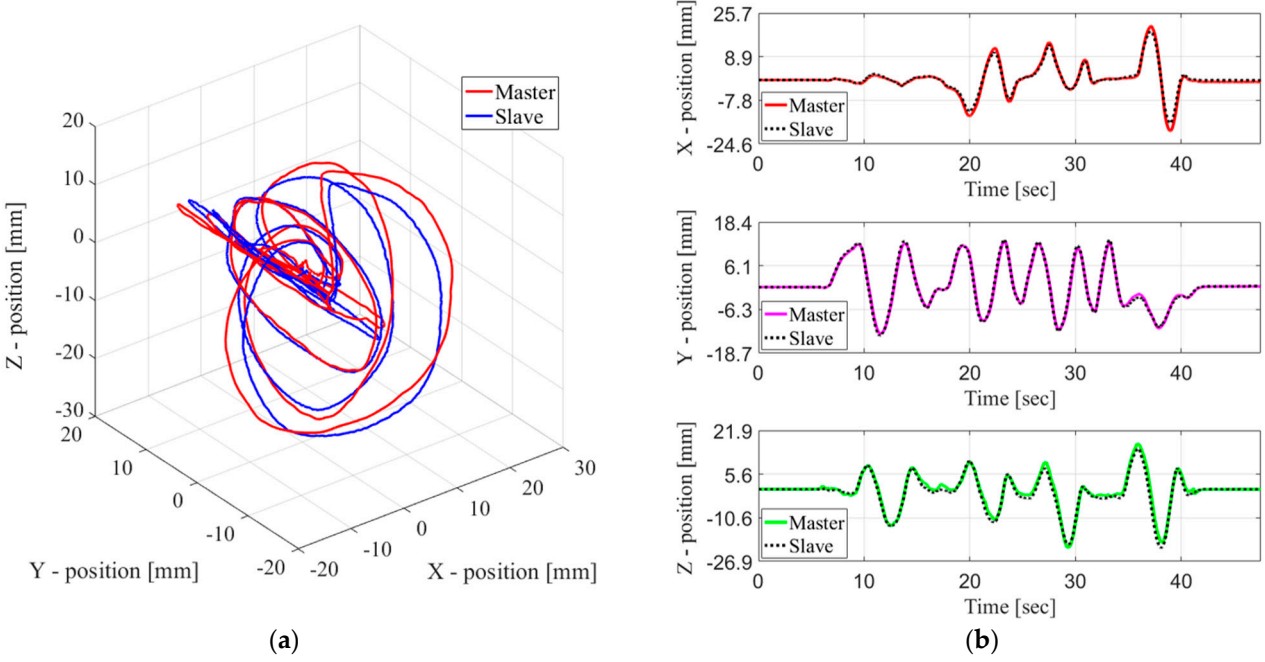

(a)                                                                                                                          (b)

**Figure 10.** Result of the master–slave control in the developed teleoperation system: (**a**) in 3D space; (**b**) in 2D space $X$, $Y$, and $Z$ axis.

**Table 4.** Measured error of the EE on the teleoperation.

| Parameter | Mean (mm) | SD (mm) | RMS (mm) |
|---|---|---|---|
| $X$ axis | 0.2191 | 0.7625 | 0.7933 |
| $Y$ axis | 0.0284 | 0.4276 | 0.4285 |
| $Z$ axis | −0.5852 | 0.9966 | 1.1556 |
| Magnitude | 0.6255 | 1.3257 | 1.4657 |

Figure 11 shows a snapshot of the teleoperation system experiment conducted by the master and slave device through the admittance control. The red dotted lines in each figure represent the initial reference line of the slave CDPR device end-effector, and some of the paths are shown in Figure 10. Green dotted lines in each figure represent the initial reference line of the master CDPR haptic device end-effector. As a result, the master CDPR haptic device on the user's desk was able to control the slave CDPR according to the user-entered motion with visual supervising of an operator.

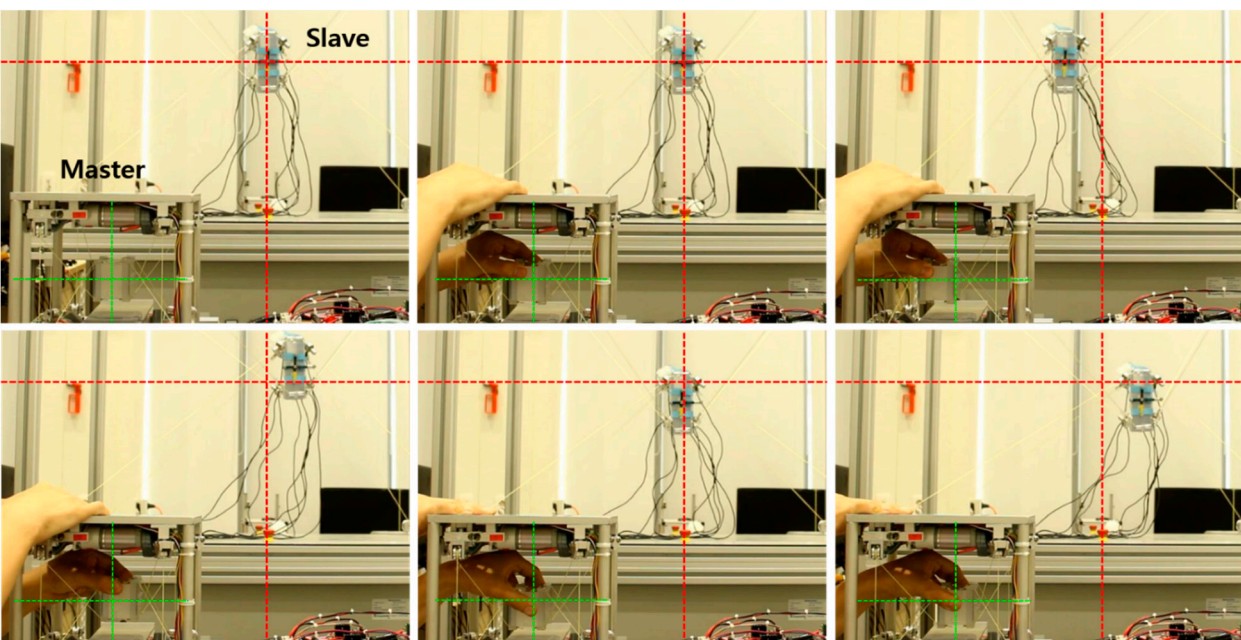

**Figure 11.** Snapshot of the master–slave teleoperation system experiment.

## 5. Conclusions and Future Works

In this paper, we presented an intuitive CDPR haptic device for user-friendly teleoperation of a fully constrained CDPR at a remote site. We designed the main hardware of the haptic device configured with the CDPR mechanism and built and embedded a controller with driving circuits based on a commercialized microprocessor. For the force control related to the reactive feeling of an operator at the master device, an adjustable admittance control scheme was applied to the control algorithm, where we considered the specifications of existing haptic devices and angle of force measurements for reducing the measurement error of the force sensors.

For the performance verification, we configured the remote-control environment by using the proposed CDPR haptic device and MINI IPAnema 3. As a result, we verified that the developed master device successfully controlled the remotely connected slave CDPR by an operator. Synchronized movements of the master and the slave device were realized, and adjustable force depending on the object was achieved. Based on the prototyped system, we validated the performance and the advantages of the proposed system for a specific task. The main contributions of this work are: (1) we advanced the conceptual intuitive bilateral teleoperation concept of using two CDPRs in [19] to be able to be used at a user's desk by prototyping a mini CDPR; and (2) the engineering parts required to build a microprocessor-based CDPR are provided in detail, which may be helpful for further relevant research into CDPR development with minimally sized parallel robots. This study shows that the microprocessor can compute an amount of complex numerical computations in a short time.

Even with the contribution of developing a small-sized haptic device capable of on-desk operation, this study has several limitations. First, in this proposed unilateral teleoperation system, the operator cannot feel the reaction force generated from the slave device when it makes contact with an obstacle in the workspace. Therefore, bilateral teleoperation using force feedback will be incorporated in our future work. Second, position errors caused by pulley size was not considered when calculating the inverse kinematics of the CDPR. This can directly affect the positional accuracy since the size of the pulley is comparably large in a small-scale CDPR, which can be assumed as negligible in a large scale CDPR. Finally, if the proposed haptic CDPR is limited to a CDPR-specific

unique device, then further study to implement it in general-purposed haptic devices will need to be carried out. We would like to consider this effect in our future work.

**Author Contributions:** Conceptualization, C.-S.K. and E.-S.K.; investigation, E.C.; methodology, M.-C.K., J.-H.P., R.B., and S.A.G.; validation, J.-H.P., M.-C.K., and E.C.; system analysis—software, M.-C.K., R.B., and J.-H.P.; formal analysis, M.-C.K., S.A.G., and E.-S.K.; writing—draft preparation, J.-H.P. and M.-C.K.; writing—review and editing, C.-S.K.; supervision, C.-S.K.; funding acquisition, J.-O.P. All authors have read and agreed to the published version of the manuscript.

**Funding:** This research was supported by a grant from the Korea Health Technology Development R&D Project through the Korea Health Industry Development Institute (KHIDI), funded by the Ministry of Health and Welfare, Korea (grant number: HI19C0642).

**Institutional Review Board Statement:** No institutional board review was necessary for this project.

**Informed Consent Statement:** Not applicable.

**Data Availability Statement:** Data sharing not applicable.

**Conflicts of Interest:** The authors declare no conflict of interest.

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
