# Peer review of "A Portable Intuitive Haptic Device on a Desk for User-Friendly Teleoperation of a Cable-Driven Parallel Robot"

_applsci, doi:10.3390/app11093823_

Round 1
Reviewer 1 Report
This paper presents a CDPR haptic device for user-friendly teleoperation.
The paper is well structured and I find the experiments are sufficients and support the conclusions.
To be acceptable for publication I suggest to improve the quality of the images. Many of them have low resolution.
I would also suggest that the authors put more emphasis on what distinguishes their work from previous ones.
Reviewer 2 Report
The paper’s topic:
The main idea of the paper is building a smaller compact-sized Cable-Driven Parallel Robot (CDPR) used as a haptic device for the teleoperation of bigger Cable-Driven Parallel Robot, admittance control (motions that result from a force input) is used for the haptic device and master slave configuration is applied.
Research question:
The intuitive and user-friendly remote control has not been addressed yet owing to the unfamiliar multiple-cable configuration of CDPRs.
Equipments:
A developed haptic device and MINI IPAnema3 CDPR and an Optical Tracking System that can measure and store the position and rotation of its marker
Structure of the paper:
The paper consists of 15 pages organized as follow:
- Introduction, state of art mostly focused on the Cable-Driven Parallel Robots, some references about its advantages, examples of parallel type of haptic devices and motivation of the paper.
- description of the system
- Kinematics, Dynamics, and Control of the haptic CDPR and this include mathematic modelling
- Design and Fabrication of the haptic CDPR and this include hardware and software specifications
- Experimental Setup and Results
- comparative experiment of desired and measured positions between master and slave
- Comparison of the applied wrench force and desired position values
- conclusion
Contributions:
According to the authors, the contributions are:
- A new type of the compact-sized CDPR configuration haptic device for user friendly teleoperation.
- Performance verification based on real experiments.
General impression and analysis:
About the contributions:
The first contribution announced by the author consists of a new type of the compact-sized CDPR configuration haptic device for user friendly teleoperation with a small time delay. Many haptic devices for parallel robots already exist like Delta Z with even force feedback which is not included in this paper.
The second contribution about real time motion control accuracy and device’s performance is good.
The paper is correctly written even if the English expression could sometimes be improved by a native-speaking person. The ideas and works are detailed and the scientific presentation and explanations are relevant.
Conclusion
*In the conclusion the authors said ‘Compared to the existing haptic devices, the developed CDPR haptic device, which has high intuitiveness, low inertia, and a large workspace, can help operators flexibly cope with this situation on the remote site by allowing them to feel as if they are actually holding and moving the control object’. Where is the comparison?
*The part presenting hardware specification is too long.
*Building a new type of master haptic device smaller than the slave cable driven parallel robot is novel as an idea but it hard to implement and will be always unique for the specified robot.
*For scientific contribution, there is no novelty in this paper, it is almost pure engineering.
*Also the method of moving manually the master haptic device with the Optical Tracking System is limited due to the risk of occultation.
* the figures’ resolution must be improved for a better quality
The paper could be accepted after improving all these points.
Reviewer 3 Report
In this paper a compact-sized haptic device based on a Cable-Driven Parallel Robot (CDPR) mechanism for teleoperation is proposed.
The haptic device is controlled by an admittance control for stiffness adjustment and implemented in an embedded microprocessor-based controller
To validate the performance of the device, authors propose some experimental simulations involving a teleoperation experimental setup by using the prototyped portable CDPR as a master and other size CDPR (MINI IPAnema 3) as a slave robot.
Main contribution of this paper deals with the realization of the whole experimental setup.
In particular, authors propose the main hardware of the haptic device configured with the CDPR mechanism and built up and embedded controller with driving circuits based on a commercialized microprocessor.
For the force control related to the reactive feeling of an operator at the master device, adjustable admittance control scheme is applied to the control algorithm,
For the performance verification, authors have configured the remote-control environment by using the proposed CDPR haptic device and MINI IPAnema3.
Paper is organized as follows
The background of the developed haptic device, including the kinematics, dynamics, and admittance control, is introduced in Section 2.
Informations about the practical design and fabrication of the compact CDPR hardware and microprocessor-based controller, as well as the proposed control algorithm and the configuration of teleoperation are proposed in Section 3.
Section 4 provides experimental setup and results .
According to this reviewer, although theoretical or methodological contributions are minor, this paper seems to be of some interest from application point of view.
An adequate level of quality for journal publication is reached.
Author Response
We really appreciate the reviewer’s valuable time and efforts to understand the manuscript. And also, we thank you so much for this constructive and positive judgement.